# Development of an Active High-Speed 3-D Vision System [note 1]

**DOI:** 10.3390/s19071572

**Published:** 2019-04-01

**Authors:** Akio Namiki, Keitaro Shimada, Yusuke Kin, Idaku Ishii

**Affiliations:** 1Department of Mechanical Engineering, Chiba University, 1-33 Yayoi-cho, Inage-ku, Chiba 263-8522, Japan; smd6ktr8@gmail.com (K.S.); yusukekin70278965@gmail.com (Y.K.); 2Department of System Cybernetics, Hiroshima University, 1-4-1 Kagamiyama, Higashi-Hiroshima, Hiroshima 739-8527, Japan; iishii@robotics.hiroshima-u.ac.jp

**Keywords:** high-speed vision, spatial coded pattern projection, depth image sensor, active tracking

## Abstract

High-speed recognition of the shape of a target object is indispensable for robots to perform various kinds of dexterous tasks in real time. In this paper, we propose a high-speed 3-D sensing system with active target-tracking. The system consists of a high-speed camera head and a high-speed projector, which are mounted on a two-axis active vision system. By measuring a projected coded pattern, 3-D measurement at a rate of 500 fps was achieved. The measurement range was increased as a result of the active tracking, and the shape of the target was accurately observed even when it moved quickly. In addition, to obtain the position and orientation of the target, 500 fps real-time model matching was achieved.

## 1. Introduction

In recent years, depth image sensors capable of obtaining 3-D information with a single sensor unit have shown remarkable progress, and various types of depth image sensors have been mainly used in robot vision research to obtain the 3-D environment of a workspace in real time. One type of depth sensor in widespread use employs a spatial coded pattern method in which the space is represented by binary codes by projecting multiple striped patterns [1]. In this method, no special function is required for the camera and projector, and it is easy to increase the spatial resolution via a simple calculation. However, there is a drawback that it takes time to recover the 3-D shape from multiple projection images. Therefore, this method is not fast enough for real-time visual feedback, and it is also difficult to use it for dynamically changing environments.

On the other hand, 1 kHz high-speed vision systems have been studied by a number of research groups [2,3]. Such a system is effective for controlling a robot at high speed, and various applications of high-speed vision have been developed [4,5]. The performance of such a target tracking system has improved [6]. One such application is a high-speed 3-D sensing system in which a 1 kHz high-speed vision system and a high-speed projector are integrated, and this system achieved 500 Hz high-speed depth sensing [7]. However, because of the limited range of the projection, the measurement range is narrow, and it is difficult to apply this system to a robot that works over a wide area.

In this paper, we propose a 500 Hz active high-speed 3-D vision system equipped with a high-speed 3-D sensor mounted on an active vision mechanism [8]. The concept is shown in Figure 1. By controlling the orientation of the camera and tracking the center of gravity of a target, this system realizes a wide measurement range and fast and highly accurate 3-D measurement. Since object tracking is performed so that the object is kept within the imaging range, the system compensates for a large motion of the object, and the state becomes nearly stationary, enabling accurate 3-D measurement. In order to use the obtained depth information for real-time robot control, real-time model matching was performed, and the position and orientation of a target object could be extracted at 500 Hz [9]. Finally, the validity of the active high-speed 3-D vision system was verified by actual experiments.

This paper is an extension of our previous conference papers [8,9]. In those papers, a basic system configuration and a basic method were proposed. However, the system details were not fully explained and there were not enough results to evaluate its performance. We have added detailed explanations and new results in this paper.

## 2. Related Work

Three-dimensional depth sensors can acquire the 3-D shape of an object in real time more easily than conventional stereo cameras. Therefore, they are increasingly used in research on robot manipulation. In most of these sensor systems, 3-D point cloud data is obtained, and model matching is performed [10,11,12].

Concerning active measurement by projecting light patterns, many methods have been proposed [13,14]. The previous methods can be divided into one-shot measurement and multi-shot measurement. In one-shot methods, a 3-D shape can be measured with just a single projection by devising a suitable pattern to be projected. Therefore, it is easy to increase the speed [15,16,17].

On the other hand, in multi-shot methods, a 3-D image is acquired by projecting multiple patterns. These methods can obtain higher-resolution images than those of the one-shot methods; however, there is the disadvantage that they are not suitable for measurement of moving objects. Examples of multi-shot methods are the spatially coded pattern method [1,18] and the phase-shift method [19,20,21,22]. The spatially coded pattern method is easy to implement; however, in order to increase the resolution, the number of projection images needs to be increased. In the phase-shift method, high-resolution images can be obtained with three projected patterns. One problem, however, is the indeterminacy of corresponding points.

Research on increasing the speed by adopting a multi-shot method in order to measure moving subjects has been conducted. As examples of the phase-shift method, a phase-decoding method that considers movements of an object [23,24] and a method that integrates the phase-shift method with Gray-code pattern projection [25] have been proposed. Hyun and co-workers have proposed a system using a mechanical projector [26]. Lohry have proposed a system using 1-bit dithered patterns [27]. An example of the spatially coded pattern method is a 500 Hz measurement system using high-speed vision [7]. In order to measure a moving object, a method in which it is assumed that the movement is in one direction [28] and a method in which the movement of an object is constrained in advance [29] have been proposed. Although these previous methods have realized sufficient speed increases, they have difficulty in measuring a target that moves quickly in a large area. In the present study, by introducing a visual servoing technique, we have solved this problem, and we propose a method suitable for robot control [8,9].

Our group have been conducting research on high-speed vision systems [2,3]. These high-speed vision systems are based on a high-frame-rate image sensor and a parallel processing unit. By utilizing parallel computing techniques, these systems can achieve real-time image processing at rates of 500∼1000 Hz or higher. This is effective for controlling a robot at high speed, and various applications of high-speed vision have been developed [4,5]. Examples include a ball juggling robot [30] and a *kendama* robot [31]. In particular, the *kemdama* robot needs to handle an object with a complicated shape and measure its 3-D shape accurately. However, accurate 3-D measurement is difficult when using only 2-D stereo vision. In this paper, therefore, we propose a system that can quickly measure the 3-D shape of a target so as to achieve dexterous quick manipulation [9].

## 3. System

The configuration of the developed system is shown in Figure 2. A camera and a projector are mounted on a two-axis active vision system. First, pattern projection and image capturing are conducted by synchronization with a trigger from a computer for visual processing (Vision PC). Second, a spatial code, a depth map and the centroid of the object are calculated from the captured images by image processing with a Graphical Processing Unit (GPU). Third, the active vision system is controlled to track the target using a real-time control system, on the basis of the data about the centroid of the target. The real-time control system is constructed using a controller manufactured by dSPACE.

### 3.1. High-Speed Depth Sensor

The projector is a DLP Light Crafter 4500 manufactured by Texas Instruments (Dallas, TX, USA). The specifications are shown in Table 1. It projects a 9-bit Gray-code pattern of 912×1060 pixels at 1000 fps. The projected patterns are explained in Section 4.1.

The high-speed vision system is based on IDP-Express manufactured by Photron Inc. (San Diego, CA, USA) The specifications are shown in Table 2. The camera captures 8-bit monochrome, 512×512 pixel images at 1000 fps. The size of the camera head is 35 mm × 35 mm × 34 mm, and the weight is 90 g. The compact size and low weight of the camera reduce the influence of inertia.

In the Vision PC, the main processor is an Intel Xeon(R) E5-1630v4 (3.70 GHz, 4core), and the GPU is an NVIDIA Geforce GTX1080Ti.

### 3.2. Active Vision

The active vision system, called AVS-III, has two orthogonal axes (pan and tilt), which are controlled by AC servo motors. The system specifications are shown in Table 3. It is capable of high-speed movement because each axis has no backlash and a small reduction ratio owing to a belt drive. The coordinate system is shown in Figure 3.

The performance of the tracking depends on the weight of the head and the power of the motor. In previous research such as [2], only a camera head was mounted and there was no projector. Therefore, it is estimated that the tracking performance was better than our system. However, we are currently designing a new head to reduce the inertia of the system to improve the performance.

### 3.3. Overall Processing Flow

A flowchart of the processing of the entire system is shown in Figure 4. It consists of three parts: high-speed depth image sensing, target tracking, and model matching. The system tracks a fast moving object by visual servoing, and a depth image is generated at 500 Hz while controlling the direction of the visual axis of the camera. Next, the generated image is converted to point cloud data, model matching is performed with a reference model, and the position and orientation of the target are detected at 500 Hz. A detailed explanation of each step is given in the next section.

## 4. High-Speed Depth Image Sensing

The system projects a 9-bit Gray-code pattern of 912×1060 pixels at 1000 fps. The projected pattern maps are shown in Figure 5a. After projection of each pattern, a black/white reversed image called a negative is projected in order to reduce errors at the borders between white and black areas. Therefore, a total of 18 image patterns are projected, and the actual measurement rate per pattern becomes 500 Hz. The reason why only 1060 of the total of 1140 lines are used is so that the projection angle corresponds with the angle of view.

The upper left corner is a space for recognizing the most significant bit pattern map, and it is impossible to measure a depth value in this area. In this system, the spatial code is composed of 265 rows (rows 0–264), which are mapped onto the 1060 projector rows while leaving enough space between rows for image adjustment because it is necessary to modify the pattern maps owing to the inclination of the optical axes of the camera and the projector.

### 4.1. Spatial Coding

In this system, we adopt the spatially coded pattern method [1] and a Gray-code pattern [32]. The reason for using a Gray-code pattern is that the code error decreases because the Hamming distance with a Gray-code is only one. When the spatial code is *n* bits, and the *k*-th projected binary image P(x,y,k), (0≤k≤n-1), has a spatial resolution of $P_{x}\times P_{y}$ pixels, $P(x_{p},y_{p},k)$ can be expressed by the following equation:(1)$$\begin{array}{c} P\left(x_{p},y_{p},k\right)=\left\lfloor \frac{y_{p}\times 2^{k}}{P_{y}}+\frac{1}{2}\right\rfloor \;\mathrm{mod}\;2, \end{array}$$
where ⌊x⌋ is the greatest integer less than or equal to *x*. To enhance the accuracy of the measurement, the projector alternately projects this pattern and its black/white inverted pattern. The projected pattern image $P(x_{p},y_{p},k)$ is called the “positive pattern”, and its inversion is called the “negative pattern”.

When a captured positive pattern image $I_{pk}\left(x_{c},y_{c}\right)$ and a captured negative image $I_{nk}\left(x_{c},y_{c}\right)$ are represented by 8-bit intensities, a binarized image $g_{k}\left(x_{c},y_{c}\right)$ is obtained by the following equation:(2)$$\begin{array}{cc} g_{k}\left(x_{c},y_{c}\right) & =\left\{\begin{array}{cc} 1 & \left(I_{pk}\left(x_{c},y_{c}\right)-I_{nk}\left(x_{c},y_{c}\right)>\psi \right)\\ 0 & \left(I_{pk}\left(x_{c},y_{c}\right)-I_{nk}\left(x_{c},y_{c}\right)<-\psi \right)\\ \phi & \mathrm{else} \end{array}\right. \end{array}$$
where ψ is an intensity threshold for binarization, and ϕ is an ambiguous point because of occlusion.

The Gray code $G(x_{c},y_{c})$ is determined from *n* binary images:(3)$$\begin{array}{c} G\left(x_{c},y_{c}\right)=\left\{g_{0}\left(x_{c},y_{c}\right),g_{1}\left(x,y\right),\cdots ,g_{n-1}\left(x_{c},y_{c}\right)\right\}. \end{array}$$

Then, $G(x_{c},y_{c})$ is converted into the binary code $B(x_{c},y_{c})$.

### 4.2. Stereo Calculation

By using the spatial code $B(x_{c},y_{c})$, the corresponding points on the projected image and the captured image can be detected. The relationship between the camera pixel coordinate, $u_{c}={\left[x_{c},y_{c}\right]}^{T}$, the projector pixel coordinate, $u_{p}={\left[x_{p},y_{p}\right]}^{T}$, and the 3-D position of the corresponding projected point, $r_{c}={\left[X_{c},Y_{c},Z_{c}\right]}^{T}$, is expressed as $h_{c}{\bar{u}}_{c}=C{\bar{r}}_{c}$ and $h_{p}{\bar{u}}_{p}=P{\bar{r}}_{c}$ where $C\in R^{3\times 4}$ and $P\in R^{3\times 4}$ are the camera parameter matrix and the projector parameter matrix obtained by calibration, and $h_{c}$ and $h_{p}$ are scale parameters. The operator $\bar{x}$ means a homogeneous vector x.

Thus, $r_{c}$ is obtained as
(4)$$\begin{array}{c} Fr_{c}=R,\;\;\;\;F≜\left[\begin{array}{ccc} C_{11}-C_{31}x_{c} & \;C_{12}-C_{32}x_{c} & \;C_{13}-C_{33}x_{c}\\ C_{21}-C_{31}y_{c} & \;C_{22}-C_{32}y_{c} & \;C_{23}-C_{33}y_{c}\\ P_{21}-P_{31}y_{p} & \;P_{22}-P_{32}y_{p} & \;P_{23}-P_{33}y_{p} \end{array}\right],\;\;\;\;R≜\left[\begin{array}{c} C_{34}x_{c}-C_{14}\\ C_{34}y_{c}-C_{24}\\ P_{34}y_{p}-P_{24} \end{array}\right] \end{array}$$

The results are calculated off-line and are used as a lookup table. For on-line measurements, using this look-up table improves execution speed.

### 4.3. Accuracy of Measurement

Consider the relationship between the frame rate and the delay. Although the frame rate of the camera and the projector is 1000 Hz, since one bit is obtained with a pair of images consisting of a positive image and its negative image in order to improve the accuracy, the frame rate for one-bit acquisition is actually 500 Hz. To generate a 9-bit depth image, 18 images are required. However, the depth image is updated every two images, that is, every 1-bit acquisition. Therefore, the update frame rate is 500 Hz, but its delay is 2×18=36 ms. This delay may degrade the accuracy of depth recognition for a moving target. However, if the target tracking is performed as described in the next section, the relative position error between the object and the camera becomes sufficiently small, and the target object can be regarded as if it has stopped.

Let us now analyze the relationship between measurement accuracy and target speed. The geometrical relationship in the active stereo vision is actually described by Equation (Equation 4). However, in order to simplify the problem, consider a simplified model shown in Figure 6, where it is assumed that the target is on a plane and the size of the projected plane is W×H, the distance between the plane and the optical center is *D*, the distance between the projector and the camera is *t*, and n=9 shows the number of the projected patterns.

Then, the maximum resolution in the horizontal direction is given by
(5)$$\begin{array}{c} \Delta y=\frac{H}{2^{n}}. \end{array}$$

In our system, the throw ratio of the projector D/W=1.2, and W/H=912/1060, t=0.2 m. Assuming that D=1.0 m, the minimum resolution in the horizontal direction is given by Δy=1.4 mm, and that in the depth direction is given by Δz≃Δy/t×D=7 mm.

The time to obtain one depth image is given by $\Delta t=2n\times 10^{-3}=18$ ms. Therefore, the maximum speed at which recognition can be performed is given by
(6)$$\begin{array}{c} v_{x}=\frac{\Delta y}{\Delta t}=7.78\;\mathrm{cm}/s,\;\;\;v_{z}=\frac{\Delta z}{\Delta t}=38.89\;\mathrm{cm}/s. \end{array}$$

Tis relative speed caused by the tracking delay of the camera can be compensated by adopting the visual tracking described in the next section. If the speed of the target is more than the speed shown above, the resolution will be reduced; however, its influence can be minimized by employing visual tracking.

### 4.4. Correction of Spatial Code

When only a high-order bit changes and lower bits remain unchanged, there is a possibility that the spatial code changes too much. Consider an example of a 4-bit Gray code, as shown in the Table 4. When 0010=3 is obtained at step *k* and the most significant bit (MSB) changes to 1 at step k+1, the gray code become 1010=12 because the lower bits in step *k* are used. This change is obviously too large compared to the actual change. In this case, the boundary is where the MSB switches between 0100=7 and 1100=8, and the farther from this boundary, the more the code changes. Therefore, we introduced the following correction method. A change of the MSB is suppressed when the code is away from the boundary. When the code is close to the boundary, the lower bits are forcibly set to 100⋯0 to prevent a large change. The correction is executed in not only the MSB but also the lower bits. The process is described as follows (Algorithm 1).
**Algorithm 1:** Correction of spatial code.  **if**
$g_{k}\left(i,j\right)$ is changed to 0→1 or 1→0
**then**   $g_{k+1}\left(i,j\right)\to 1$
   $g_{l}\left(i,j\right)\to 0\;\;\;\;\left(l=k+2,k+3,\ldots \right)$  **end if**

## 5. Visual Tracking

### 5.1. Image Moment for Tracking

In our system, the direction of the camera–projector system is controlled according to the motion of the target. The desired direction is calculated by an image moment feature obtained from 3-D depth information. The (k+l)th image moment *m* is given by
(7)$$m_{kl}=\sum_{i}\sum_{j}A\left(x_{c_{i}},y_{c_{j}}\right){x_{c_{i}}}^{k}{y_{c_{j}}}^{l},$$
where $A(x_{c_{i}},y_{c_{j}})$ is a binary image of the target at the pixel $\left(x_{c_{i}},x_{c_{j}}\right)$ on the camera image and is given by
(8)$$A\left(x_{c_{i}},y_{c_{j}}\right)=\left\{\begin{array}{cc} 1 & \left(\phi_{l}<h_{c}\left(x_{c_{i}},y_{c_{j}}\right)<\phi_{h}\right)\\ 0 & \mathrm{else}, \end{array}\right.$$
where $\phi_{l}$ and $\phi_{h}$ are thresholds that show the depth range of the target object.

The image centroid $m_{g}$ is calculated as
(9)$$m_{g}={\left[\frac{m_{10}}{m_{00}}\;\frac{m_{01}}{m_{00}}\right]}^{T},$$
and $m_{g}$ is used as a desired position for target tracking control.

### 5.2. Visual Servoing Control

To realize a quick response, we adopt image-based visual servoing control [33]. In our system, the mechanism has only two degrees-of-freedom (2-DOF) joints, and the control method can be simplified as shown in [2,4]. The kinematic parameters are shown in Figure 3.

Assuming a pinhole camera model, the perspective projection between the image centroid $m_{g}$ and the 3-D position of the target $r_{c}={\left[X_{c}\;Y_{c}\;Z_{c}\right]}^{T}$ in the camera coordinates can be expressed as
(10)$$\begin{array}{c} m_{g}={\left[-\frac{fZ_{c}}{X_{c}},\frac{fY_{c}}{X_{c}}\right]}^{T}, \end{array}$$
where *f* is the focal length. Differentiating both sides yields
(11)$$\begin{array}{c} {\dot{m}}_{g}=\left[\begin{array}{ccc} \frac{fZ_{c}}{{\left(X_{c}\right)}^{2}} & 0 & -\frac{f}{X_{c}}\\ -\frac{fy_{c}}{{\left(X_{c}\right)}^{2}} & \frac{f}{X_{c}} & 0 \end{array}\right]{\dot{r}}_{c}≃\left[\begin{array}{ccc} 0 & 0 & -\frac{f}{X_{c}}\\ 0 & \frac{f}{X_{c}} & 0 \end{array}\right]{\dot{r}}_{c}, \end{array}$$
where it is assumed that $X_{c}\gg Y_{c}$ and $X_{c}\gg Z_{c}$ when the distance between the object and the camera is large enough and when the object is kept around the image center. For image-based visual servoing, the depth data $X_{c}$ may be an estimated value and need not be an accurate value. However, this affects the control system because of the change of the control gain due to changes of $X_{c}$.

The positional relationship between r in the world coordinates and $r_{c}$ in the camera coordinates can be expressed as
(12)$$\bar{r}=T_{c}{\bar{r}}_{c},$$
where $T_{c}$ is the homogeneous transformation and the operator $\bar{□}$ shows a homogeneous vector. Differentiating both sides using joint angle vector $q={\left[q_{1}\;q_{2}\right]}^{T}$ yields:(13)$$\dot{\bar{r}}={\dot{T}}_{c}{\bar{r}}_{c}+T_{c}{\dot{\bar{r}}}_{c}=\left[\frac{\partial T_{c}}{\partial q_{1}}{\bar{r}}_{c}\;\frac{\partial T_{c}}{\partial q_{2}}{\bar{r}}_{c}\right]\dot{q}+T_{c}{\dot{\bar{r}}}_{c}.$$

If the movement of the object is minute, it can be considered that $\dot{\bar{r}}=0$:(14)$${\dot{\bar{r}}}_{c}=-T_{c}^{-1}\left[\frac{\partial T_{c}}{\partial q_{1}}{\bar{r}}_{c}\;\frac{\partial T_{c}}{\partial q_{2}}{\bar{r}}_{c}\right]\dot{q}.$$

The relationship between ${\dot{m}}_{g}$ and $\dot{q}$ is obtained from Equations (Equation 11) and (Equation 14):(15)$$\begin{array}{c} {\dot{m}}_{g}=J\dot{q},\;\;\;\;J≜\left[\begin{array}{cc} -\frac{f}{X_{c}}\left(\left(a+X_{c}\right)\cos q_{2}+Y_{c}\sin q_{2}\right) & 0\\ -\frac{fZ_{c}}{X_{c}}\sin q_{2} & -\frac{f}{X_{c}}\left(a+X_{c}\right) \end{array}\right]≃\left[\begin{array}{cc} -f\cos q_{2} & 0\\ 0 & -f \end{array}\right] \end{array}$$
where it is assumed that *a*, $Y_{c}$ and $Z_{c}$ are sufficiently small. The matrix *J* is called the “image Jacobian”.

The active vision system tracks the target by torque control in accordance with the block diagram in Figure 4:(16)$$\tau =K_{P}J^{-1}\left(m_{d}-m_{g}\right)-K_{v}\dot{q},$$
where $K_{P}$ is the positional feedback gain, and $K_{v}$ is the velocity feedback gain. The vector $m_{d}$ is the target position in the image, and it is usually set to ${\left[0,0\right]}^{T}$. Notice that this control algorithm is based on the assumption that the frame rate of the vision system is sufficiently high. Because the vision obtains 3-D data at high-speed rate, it is not necessary to predict the motion of the target. Therefore, the control method is given as a simple direct visual feedback.

The target tracking starts when an object appears in the field of view. With regard to the model matching part, since it takes time for the initial processing, there is a possibility that the matching result may not be stabilized if the movement of the object is fast. On the other hand, the parts involving tracking and 3-D measurement work stably.

## 6. Realtime Model Matching

In this system, 3-D depth information can be output at high speed, but the data size is too large for real-time robot control. Therefore, the system is designed to output the target position and orientation by model matching [9]. The object model is given in the form of point cloud data (PCD), and the sensor information is also point cloud data. In this paper, we call the model point cloud data M-PCD, and the sensor point cloud data S-PCD. First, initial alignment is made, and then the position of the final model is determined by using the Iterative Closest Point (ICP) algorithm.

### 6.1. Initial Alignment

At each starting point of target tracking, initial alignment is performed by using a Local Reference Frame (LRF) [34]. For each point in the S-PCD and the M-PCD, the covariance matrix and its eigenvectors, as well as the normal of the curved surface, are calculated. Let *N* be the total number of points in the S-PCD and M-PCD, and $p_{i}$(i=1,2,…,N) be the position of each point in the camera coordinate system. The center $c_{i}$ and the covariance matrix $C_{i}$ of the neighborhood points of $p_{i}$ can be obtained by the following expressions [35]:(17)$$\begin{array}{cc} c_{i} & =\frac{1}{n}\sum_{j=1}^{n}q_{j} \end{array}$$
(18)$$\begin{array}{cc} C_{i} & =\frac{1}{n}\sum_{j=1}^{n}\left[\left(q_{j}-c_{i}\right){\left(q_{j}-c_{i}\right)}^{T}\right], \end{array}$$
where *n* is the total number of neighbors of $p_{i}$, and $q_{j}$ are the coordinates of the neighborhood of $p_{i}$. Let $\lambda_{0},\lambda_{1},\lambda_{2}$
$\left(\lambda_{0}\leq \lambda_{1}\leq \lambda_{2}\right)$ be the eigenvalues of $C_{i}$, and $e_{0},e_{1},e_{2}$ be the corresponding eigenvectors. At this time, $e_{0}$ coincides with the normal direction of the curved surface. Since the normal here is obtained from discrete neighbor points, it is actually an approximation of the normal.

It is necessary to correct the direction of the normal vector. The angle between the vectors $p_{i}$ and $e_{0}$ is given by $\theta_{i}=\cos^{-1}\left(\frac{p_{i}\cdot e_{0}}{∥p_{i}∥\times ∥e_{0}∥}\right)$. If $\theta_{i}<\frac{\pi }{2}$, the direction of $e_{0},e_{1}$ is reversed.

Next, points with large curvature are selected as feature points of the point clouds. The curvature is obtained by
(19)$$\begin{array}{cc} k_{i} & =\frac{2d_{i}}{\mu_{i}^{2}}, \end{array}$$
where $d_{i}=\left|\right|e_{0}\cdot \left(p_{i}-c_{i}\right)\left|\right|$, $\mu_{i}=\frac{1}{n}\sum_{j=1}^{n}∥q_{j}-p_{i}∥$. In order to find a feature point, the following normalized curvature $k_{i}^{\prime }=\frac{k_{i}}{k_{max}}$ is actually used instead of $k_{i}$: where $k_{max}$ is the maximum value of $k_{i}\left(i=1,2,\ldots ,N\right)$.

Then, LRFs are generated at the selected feature points. In each LRF, the three coordinate axes are basically given as the corrected normal vector, the second eigenvector, and their outer product. For each feature point in the S-PCD and one feature point in the M-PCD, if the target object is moved and rotated so that the directions of the LRF coincide, and if feature points are selected in the same place, the target object and the model object match. However, since the M-PCD and the S-PCD are discrete data, it is not always possible to obtain feature points in the same place. To achieve robust matching, the second axis is replaced to use a voting method from the direction of the normal around the characteristic point [36]. Also, when the point cloud is symmetric when obtaining the second axis, two candidates of the second axis are detected. Therefore, the matching is performed for two kinds of second axes.

The homogeneous transformation *T* of the target object when matching an LRF of the M-PCD with an LRF of the S-PCD can be obtained by the following expression:(20)$$T\left[\begin{array}{cccc} l_{0} & l_{1} & l_{2} & p\\ 0 & 0 & 0 & 1 \end{array}\right]=\left[\begin{array}{cccc} l_{0}^{\prime } & l_{1}^{\prime } & l_{2}^{\prime } & q\\ 0 & 0 & 0 & 1 \end{array}\right]$$
where q is the position and $l_{0}^{\prime }$, $l_{1}^{\prime }$, and $l_{2}^{\prime }$ are the axes of the LRF of the M-PCD, and p is the position and $l_{0}$, $l_{1}$, $l_{2}$ are the axes of the LRF of the S-PCD.

To select the most suitable LRF, a threshold is set in advance, and only LRFs having a curvature exceeding the threshold are selected. Furthermore, the LRFs are down-sampled at a certain rate, which is set to 0.1 in this system. The sum of the distances between points in the S-PCD and points in the M-PCD is calculated for all combinations, and the LRF that minimizes the distance is selected.

### 6.2. ICP Algorithm

After the initial alignment, precise alignment is performed using the ICP algorithm [37]. In view of processing time and stability, we adopt the Point to Point method. ICP is a popular method in 3-D matching, so we briefly explain the processing flow in this section.

Let $N_{p}$ be the number of points in S-PCD *P* and $N_{x}$ be the number of points in M-PCD *X*. The closest point among *X* is associated with each data point p belonging to *P*. To speed up this association, we use a kd-tree to store the elements of the model. A point of *X* closest to p is determined as follows:(21)$$y=\arg\min_{x\in X}∥x-p∥,$$
and let Y=C(P,X) be the set of y by using the operator C for obtaining the nearest neighbor point.

Next, to calculate the correlation of the point y in *X* closest to p, a covariance matrix is given by
(22)$$\Sigma_{py}=\frac{1}{N_{p}}\sum_{i=1}^{N_{p}}\left[\left(p_{i}-\mu_{p}\right){\left(y_{i}-\mu_{y}\right)}^{T}\right],$$
where
(23)$$\mu_{p}=\frac{1}{N_{p}}\sum_{i=1}^{N_{p}}p_{i},\mu_{y}=\frac{1}{N_{p}}\sum_{i=1}^{N_{p}}y_{i}.$$

The singular value decomposition of the covariance matrix obtained by Equation (Equation 22) is:(24)$$\Sigma_{py}=U_{py}S_{py}{V_{py}}^{T}.$$

Using $U_{py}$ and $V_{py}$, the optimum rotation *R* can be expressed by the following equation [38]:(25)$$R=U_{py}\left[\begin{array}{ccc} 1 & 0 & 0\\ 0 & 1 & 0\\ 0 & 1 & \det(U_{py}{V_{py}}^{T}) \end{array}\right]{V_{py}}^{T}.$$

The optimal translation vector t is obtained from this rotation matrix and the center of gravity:(26)$$t=\mu_{y}-R\mu_{p}.$$

In the real-time model matching, the Equations (Equation 21), (Equation 25) and (Equation 26) are iteratively performed in one step. To speed up the convergence, the iterative calculation is divided into three stages. It is performed $n_{1}$ times for samples thinned out to a quarter of the S-PCD, $n_{2}$ times for samples thinned out to half, and $n_{3}$ times for all points. Hereinafter, the iteration count is described as $\left[n_{1},n_{2},n_{3}\right]$ times. For this time, convergence judgment based on the squared error between point groups is not performed, and the number of iterations is fixed.

Notice that the difference between the current S-PCD and the previous S-PCD is small because the rate of 3-D sensing is 500 Hz. For this reason, ICP can converge with a sufficiently small number of steps.

### 6.3. Down-Sampling

Although the high-speed 3-D sensor can obtain 3-D values with all 512×512 pixels, the processing time becomes too large if all of them are transformed to PCD. Therefore, it is necessary to reduce the number of points in the PCD.

First, the number $i_{number}$ of target points is set. Next, the ratio $C_{p}$ with respect to the target PCD number is calculated as follows using the number of pixels *M* from which the 3-D values can be obtained:(27)$$\begin{array}{c} C_{p}=\left\{\begin{array}{cc} \frac{M}{i_{number}} & \mathrm{if}\;M>i_{number}\\ 1 & \mathrm{else} \end{array}\right. \end{array}$$

For i=1,2,3…, only the $i\times C_{p}$th pixel ($i\times C_{p}<512\times 512$) is point clustered. Thereafter, down-sampling is performed to eliminate points within a cube of one side length *r* having the respective centers of gravity, and outlier points are removed when the average of 20 neighboring points is more than a certain value. Down-sampling is also performed for the M-PCD as well. In the experiments described in the next session, the final number of points was about 410.

## 7. Experiment

A video of the experimental results described in this section can be seen in [39].

### 7.1. High-Speed Depth Image Sensing

In the experiment, as the target object we used a “*ken*”, which is a *kendama* stick, because it is composed of a complex curved surface, as shown in Figure 7b. In order to confirm that the sensor system could make 3-D measurements and could track the target, the *ken* at the tip of a rod was moved in front of the sensor. In this experiment, only 3-D measurement was performed without the real-time model matching.

#### 7.1.1. Result with Target Tracking Control

In this section, we explain a result of 3-D measurement shown in [8]. The image centroid is shown in Figure 8. The system and the depth map expressed in color according to the 3-D value from the camera are shown in Figure 9. The *ken* is shown in the left image, and the depth map from the 3-D sensor is shown in the right image. The color of the depth map becomes redder as the object comes closer to the camera and bluer as the object goes farther away from the camera.

The time taken to calculate the depth value was about 1.5 ms. This is shorter than the 2 ms taken to project 1-bit positive and negative pattern images. As seen in the depth map data in Figure 9, 3-D measurement seems to have been successful on the whole. Moreover, the sensor system could keep the target object within the field of view of the camera even when the *ken* moved dynamically, indicating that tracking of a moving object was also successful. However, when the *ken* moved violently, for example, at 25.0 s, the center of the image and the centroid of the *ken* were separated. This was due to the control method because there was no compensation for gravity and inertia. This issue will be addressed in future work. Regarding the depth maps in Figure 9, not only did the *ken* shape appear in the image, but also the inclination in the depth direction of the *ken* can be seen by the color gradation.

The temporal response for the position of the centroid of the object is shown in Figure 10a. The values in Figure 10a are expressed in the world coordinates. The 3-D trajectory of the center of gravity of the target is shown in Figure 10.

The trajectory of the centroid in Figure 10 was calculated offline by using the same model matching method. This is just to show the movement of the target for reference. Considering the size of the *ken* in the image and the trajectory, a wider 3-D measurement range could be achieved by tracking the target instead of fixing the high-speed 3-D sensor.

#### 7.1.2. Result without Tracking Tracking Control

For comparison, Figure 11 shows the depth map when target tracking was not performed. It can be seen that the grip of the *ken* disappeared when it was swung largely in the lateral direction. This is probably because the 3-D measurement did not catch up with the moving speed of the *ken*. On the other hand, when tracking was performed, such disappearance did not occur. This is probably because the relative target movement speed was reduced by target tracking.

### 7.2. Model Matching

#### 7.2.1. *Ken*

An experiment for the model matching was conducted using the *ken*. It was attached to a 7-DOF robot manipulator, and the true values of its position and orientation were calculated from the joint angles of the manipulator. These were compared with the values measured with the high-speed depth image sensor. The coordinate system fixed to the *ken* is shown in Figure 12a. Figure 12b shows the experimental setup of the manipulator and the high-speed depth image sensor. Figure 12c shows the M-PCD of the *ken*. The motion of the manipulator was such that it rotated about $\frac{\pi }{2}$ rad around the Z axis, $\frac{\pi }{4}$ rad around the Y axis, and $-\frac{\pi }{4}$ around the Z axis during about 2 s. In this trajectory, the maximum speed of the *ken* was about 1 m/s.

In the ICP, the number of iterations was set such that $\left[0,0,1\right]$, $i_{number}=250$, and r=7.57 mm. We set the number of points in M-PCD made from the CAD model by downsampling to 302.

The initial alignment was completed during a preparatory operation before the manipulator started to move. The time required for the initial alignment was 148 ms.

Figure 13a shows the temporal change of the number points in in S-PCD. It was adjusted depending on the parameter $i_{number}$, and it is shown that it was kept between 50 and 200 points.

Figure 13b shows the temporal change of the total processing time. It is shown that the processing time was about 1 ms and less than 1.5 ms. This means that the measurement could be achieved at a frame rate of 500 fps. In most of the measurement period, the observed S-PCD was a part of the overall body of the *ken*. However, by using the model matching, its position and orientation were accurately obtained within 2 ms.

Figure 14 shows how the M-PCD and the S-PCD match. It can be seen that the M-PCD followed the motion of the *ken*. Figure 15 and Figure 16 show the centroid and the orientation measured by the high-speed depth sensor and their true values calculated from the manipulator, respectively, in world coordinates. It is shown that there were no large errors in these measurements. It was difficult to distinguish between a large plate and a small plate in the *ken*, and it was reversed 180 degrees around the Y-axis sometimes. However, in most cases, the model matching was successful.

Notice that the cost of ICP is usually high, and it is not suitable for real-time high-speed matching in a conventional system. In our system, however, it is possible to reduce the number of iterations because the changes are sufficiently small as a result of the high frame rate.

#### 7.2.2. Cube

As a target, we used a 10 cm cube and comparing the results with the above results. Because the cube has a simpler shape than the *ken*, the model matching becomes more difficult. Therefore, we set the number of points in the M-PCD to 488 and $i_{number}$ to 350 so as to increase the number of points.

The time required for the initial alignment was 93.98 ms. The frequency of the ICP was 487 fps, which did not match the frame rate of 500 fps. Therefore, the frame rate of the ICP was asynchronous with the rate of the measurements in this experiment.

Figure 17 and Figure 18 show the results of model matching for the cube and the time response of its centroid, respectively. It can be seen that the model matching and tracking succeeded, like the result for the *ken*.

#### 7.2.3. Cylinder

We used a cylinder as a target. We set the number of points on the M-PCD to 248 and $i_{number}$ to 300. The frequency of the ICP was 495 fps.

Figure 19 and Figure 20 show the results of model matching for the cube and the time response of its centroid, respectively. It can also be seen that the model matching and tracking succeeded, like the results for the *ken* and the cube.

### 7.3. Verification of Correction of Spatial Code

We conducted an experiment to verify the effect of the correction described in Section 4.4. Figure 21 (a) shows a camera image, (b) shows the point cloud before the correction, and (c) shows the point cloud after the correction. The blue points represent the observation point cloud, and the red points represent the points to be corrected.

In (b), it can be seen that the point cloud to be corrected is in the wrong position. In (c), these points are moved around the correct position. This indicates that the proposed correction is valid.

### 7.4. Verification of Initial Alignment

In Figure 22a–d, the examples of LRFs are shown. Figure 22a shows the LRFs of the S-PCD, and Figure 22b shows the down-sampled LRFs, down-sampled to 0.1. Figure 22c,d are the original LRFs and the downsampled LRFs of the M-PCD. Figure 22e shows an example of the initial alignment. It can be seen that the M-PCD is roughly matched with the S-PCD.

## 8. Conclusions

In this paper, we proposed a 500 Hz active high-speed 3-D vision system that has a high-speed 3-D sensor mounted on an active vision mechanism. By controlling the orientation of the camera and tracking the center of gravity of a target, it realizes both a wide measurement range and highly accurate high-speed 3-D measurement. Since object tracking is performed so that the object falls within the image range, a large motion of the object is compensated for, enabling accurate 3-D measurement. Real-time model matching was also achieved, and the position and orientation of the target object could be extracted at 500 Hz.

There are some problems to be addressed in the system. First, the performance of the tracking control is not sufficient. This will be improved by adopting improved visual servoing techniques taking account of the dynamics of the system. Also, it will be necessary to decrease inertia and reduce the size of the mechanism.

Second, although the coded pattern projection method was used in our current system, multiple image projections resulted in a delay. As a way to reduce the delay, we are considering acquiring 3-D information by other methods, such as one-shot or phase-shift methods. Moreover, due to the high frame rate and the active tracking, the target motion in each measurement step is very small. By utilizing this feature, it is possible to reduce the number of projection images, and we are currently developing an improved algorithm.

## Figures and Tables

**Figure 1 sensors-19-01572-f001:**
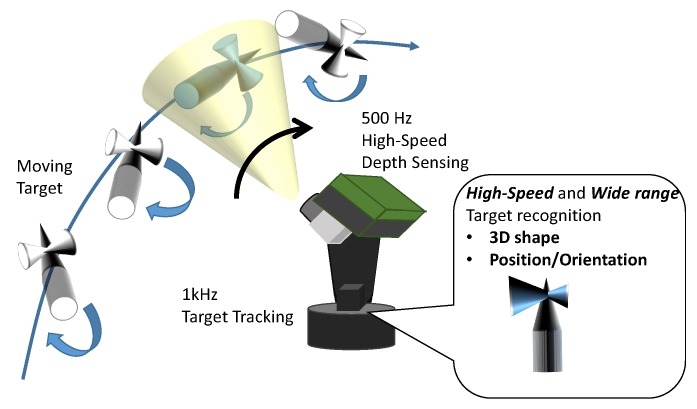
System concept.

**Figure 2 sensors-19-01572-f002:**
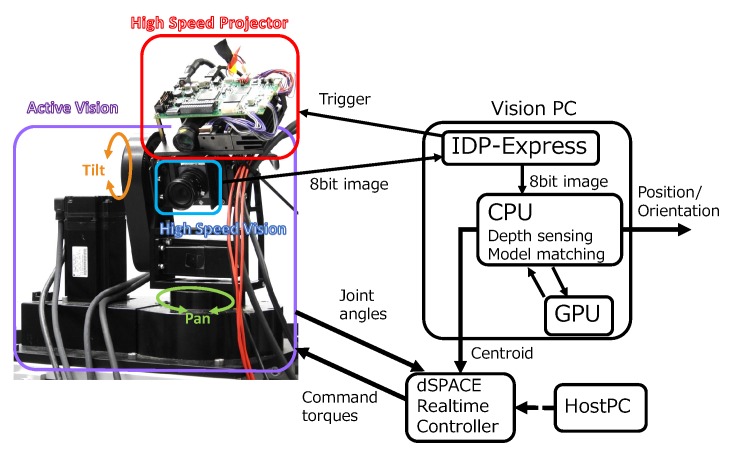
System configuration.

**Figure 3 sensors-19-01572-f003:**
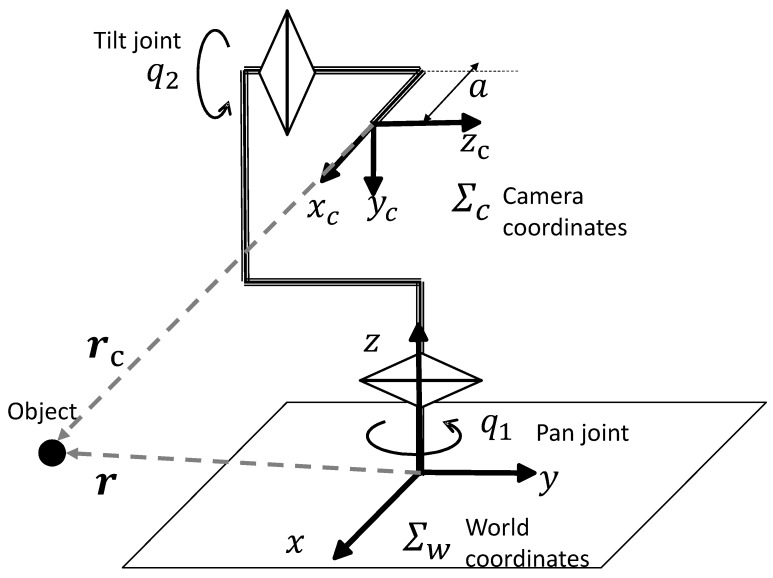
Active vision.

**Figure 4 sensors-19-01572-f004:**
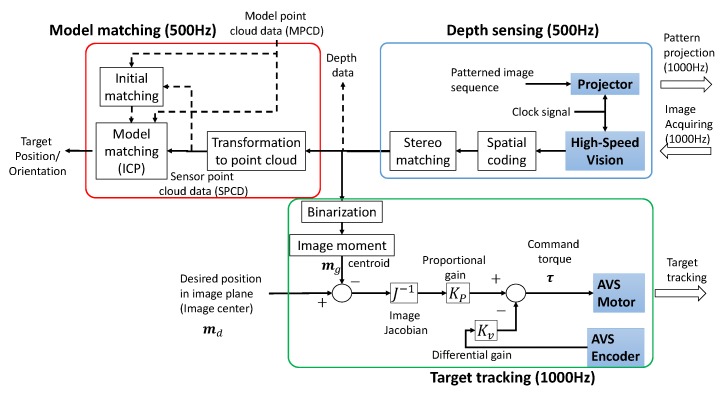
Block diagram.

**Figure 5 sensors-19-01572-f005:**
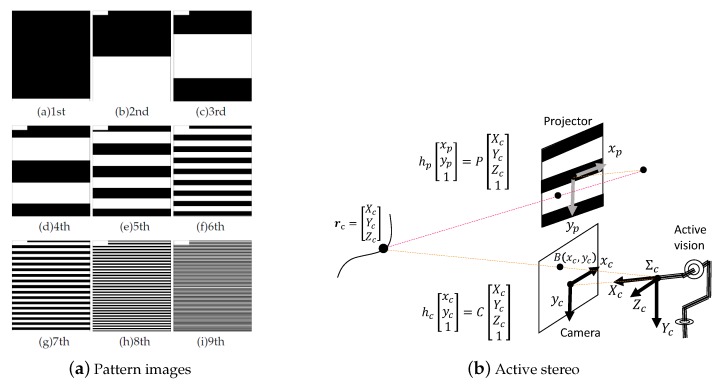
Depth sensing.

**Figure 6 sensors-19-01572-f006:**
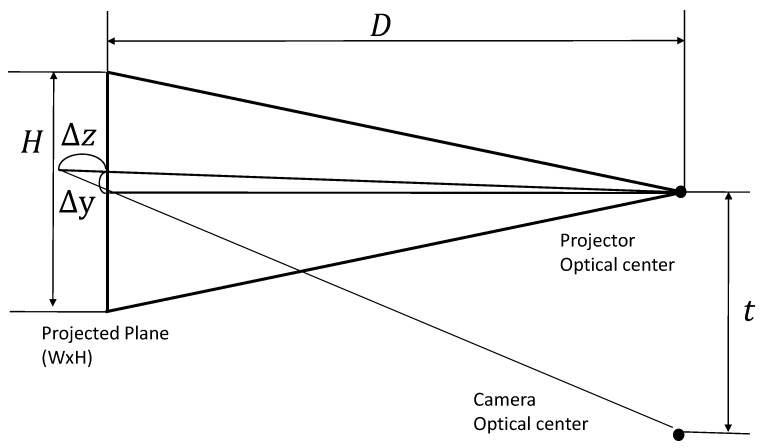
Analysis of accuracy.

**Figure 7 sensors-19-01572-f007:**
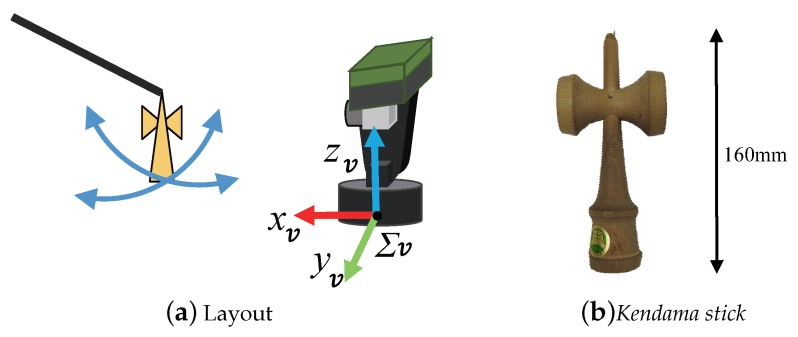
Experimental setup.

**Figure 8 sensors-19-01572-f008:**
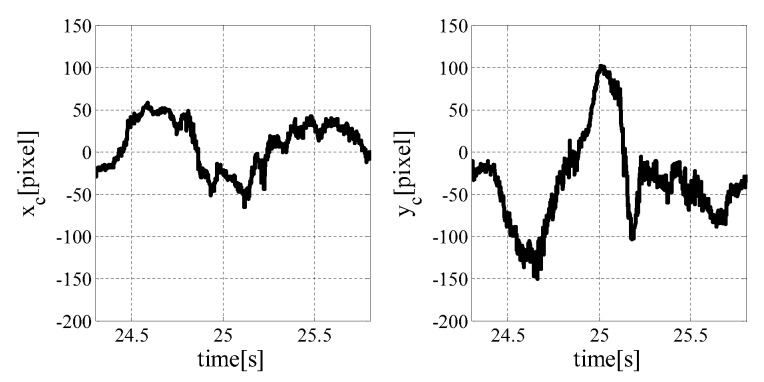
Temporal response of error in image plane [8].

**Figure 9 sensors-19-01572-f009:**
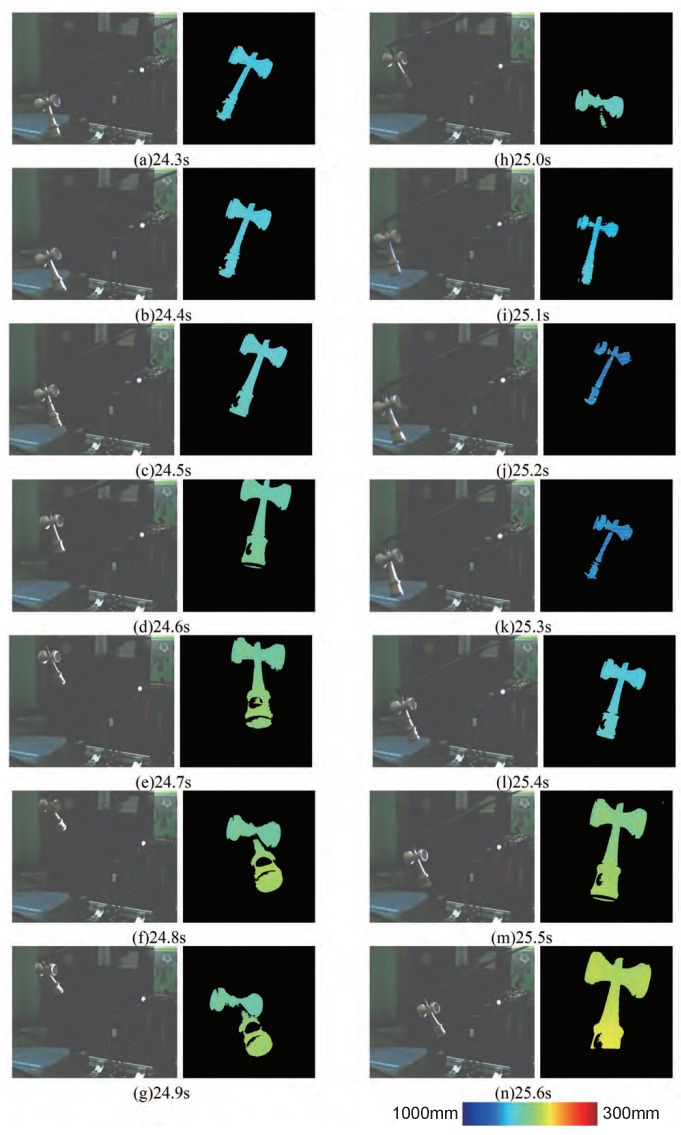
3-D measurement with target tracking [8].

**Figure 10 sensors-19-01572-f010:**
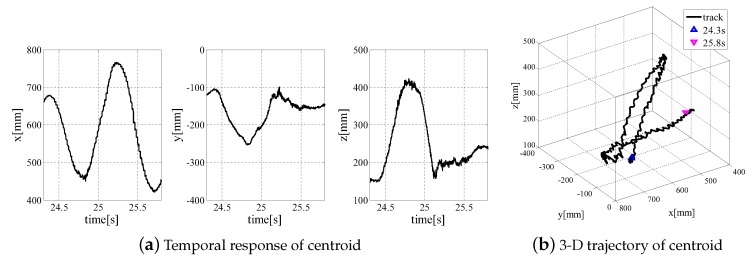
Temporal response of centroid [8].

**Figure 11 sensors-19-01572-f011:**
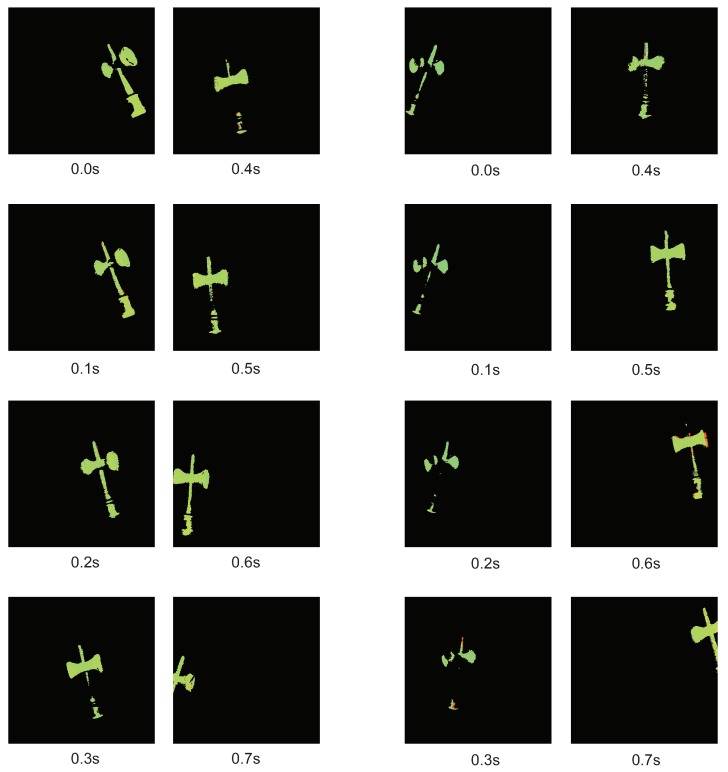
3-D measurement without target tracking.

**Figure 12 sensors-19-01572-f012:**
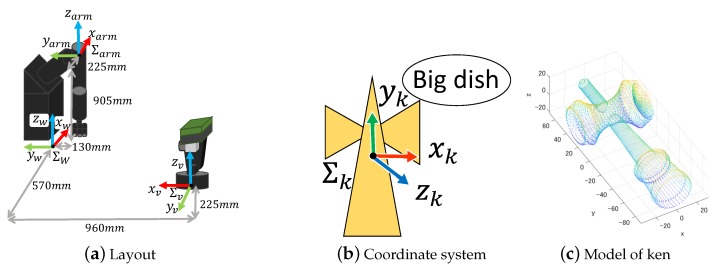
Experimental setup.

**Figure 13 sensors-19-01572-f013:**
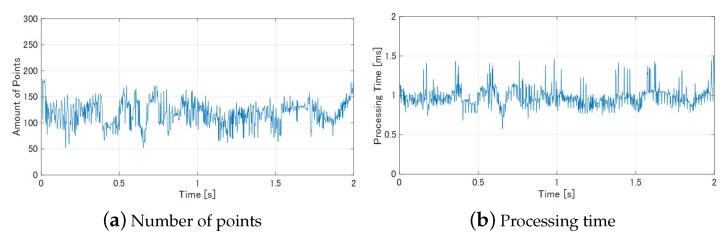
Result of model matching.

**Figure 14 sensors-19-01572-f014:**
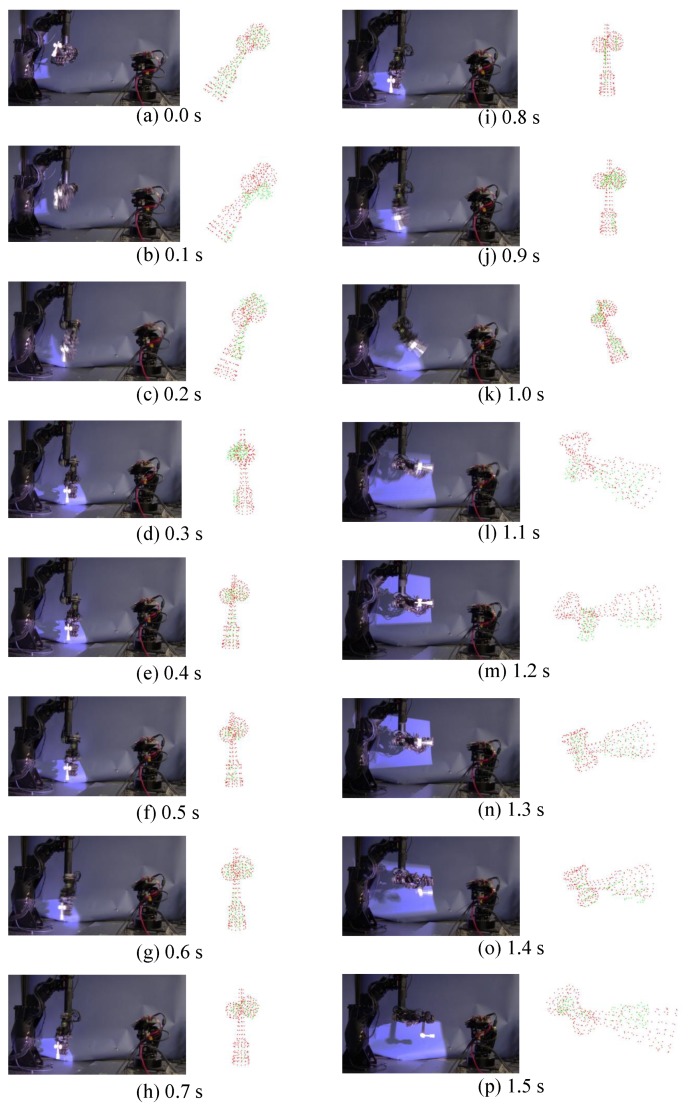
Continuous images of real-time model matching.

**Figure 15 sensors-19-01572-f015:**
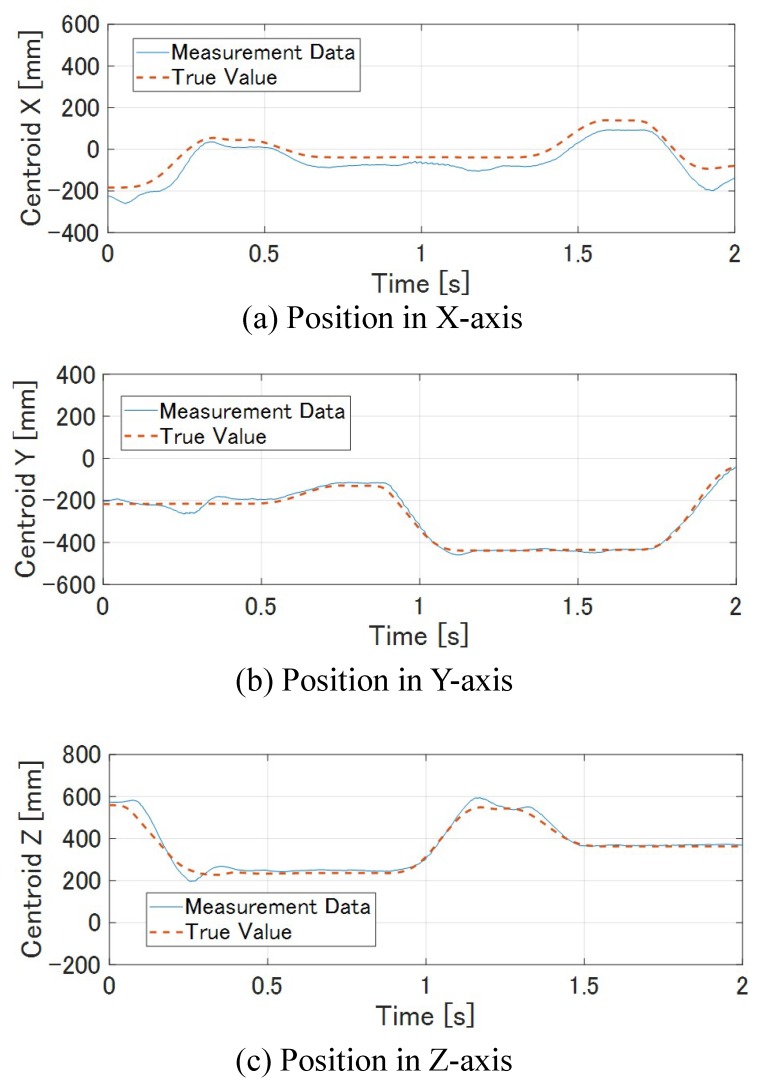
Centroid of *ken*.

**Figure 16 sensors-19-01572-f016:**
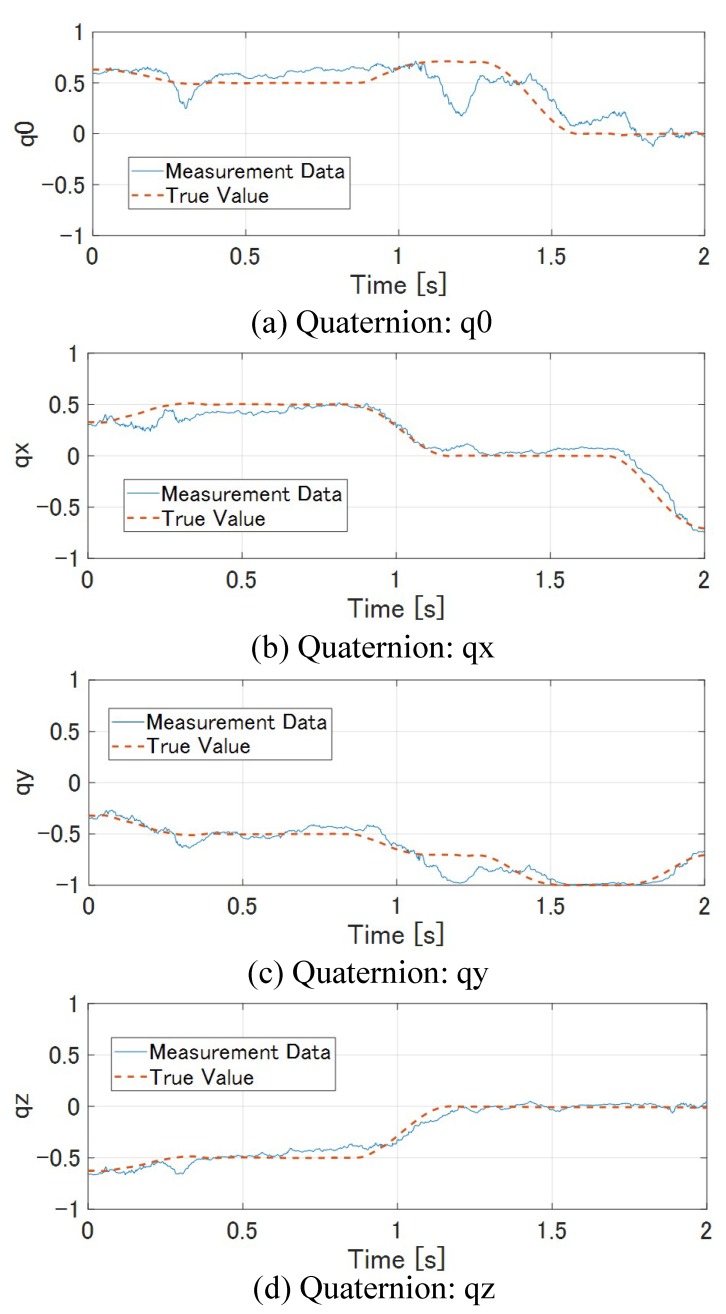
Quaternion of *ken*.

**Figure 17 sensors-19-01572-f017:**
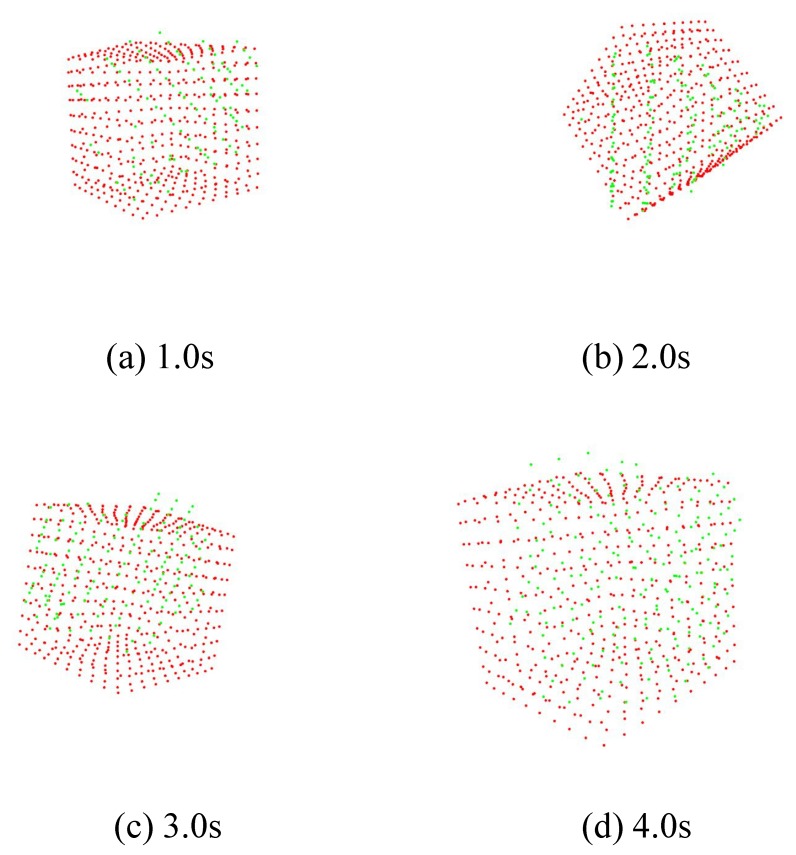
Point cloud of cube.

**Figure 18 sensors-19-01572-f018:**
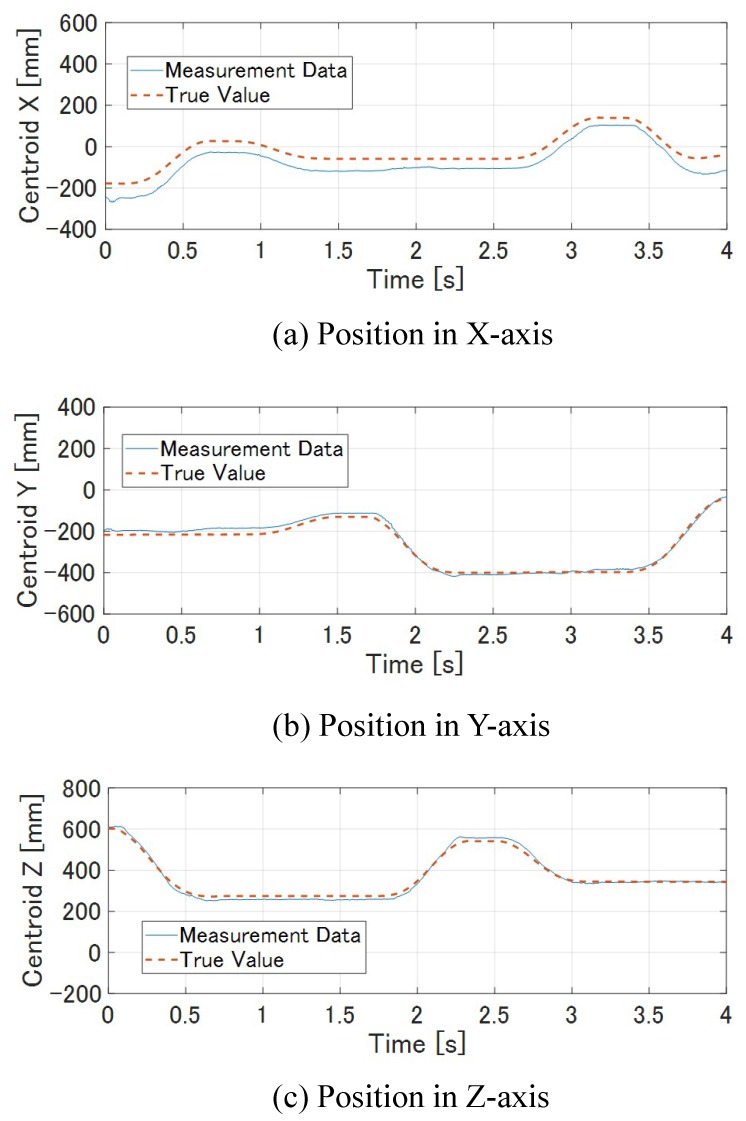
Centroid of cube.

**Figure 19 sensors-19-01572-f019:**
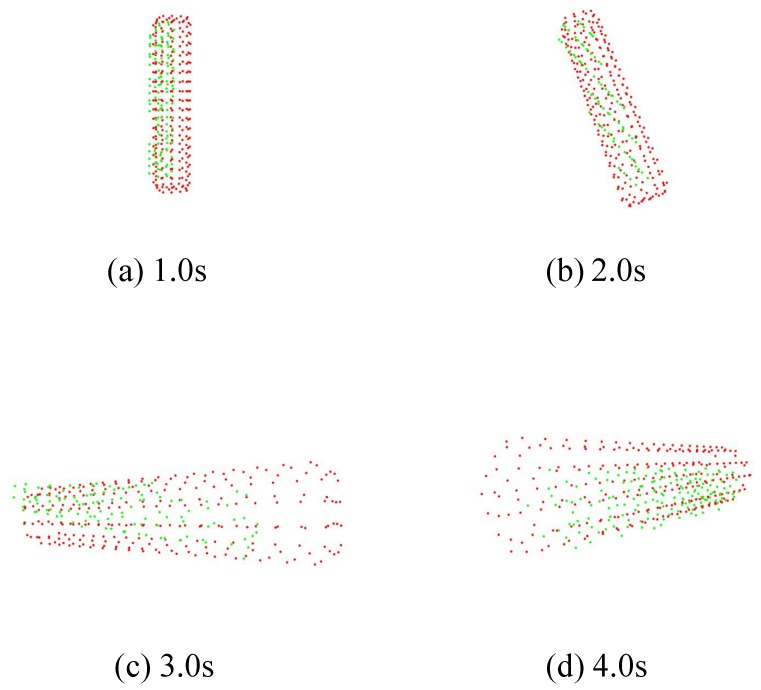
Point cloud data of cylinder.

**Figure 20 sensors-19-01572-f020:**
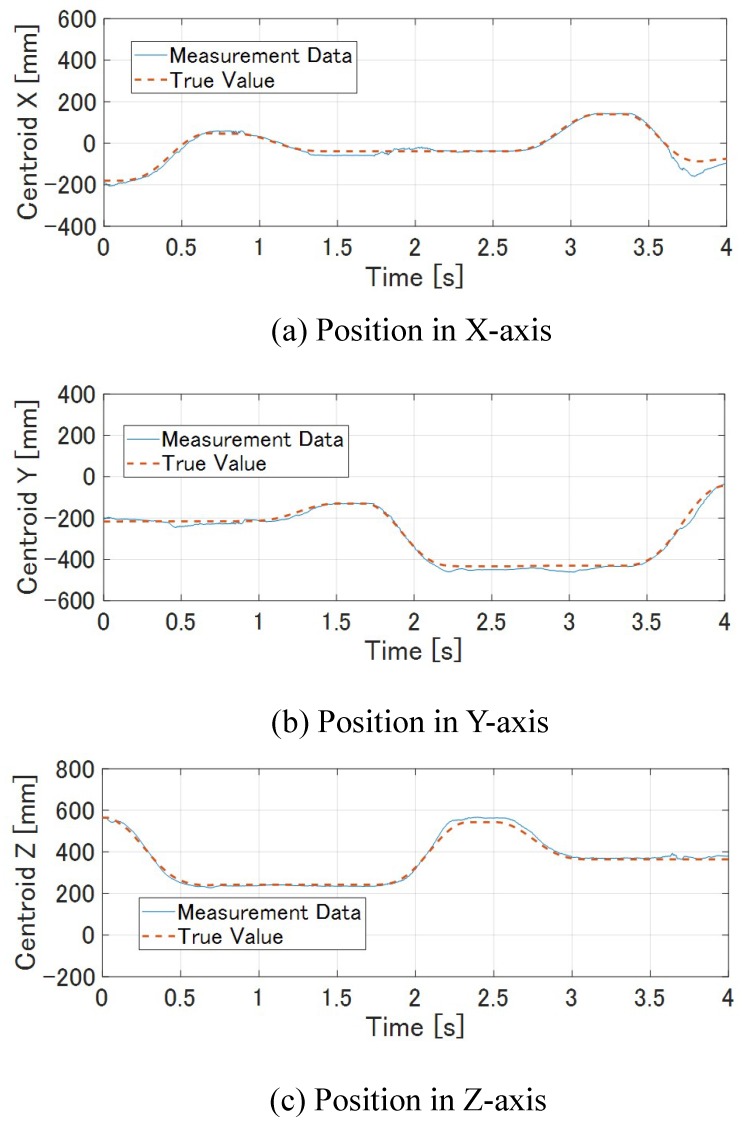
Centroid of cylinder.

**Figure 21 sensors-19-01572-f021:**
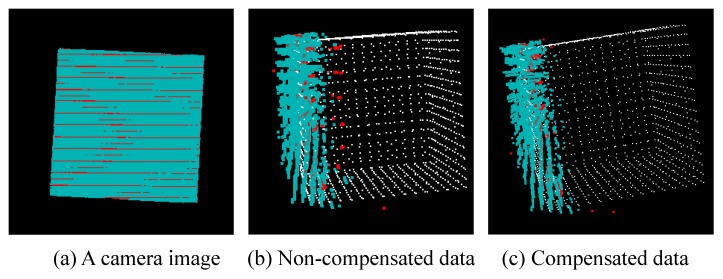
Correction of spatial code.

**Figure 22 sensors-19-01572-f022:**
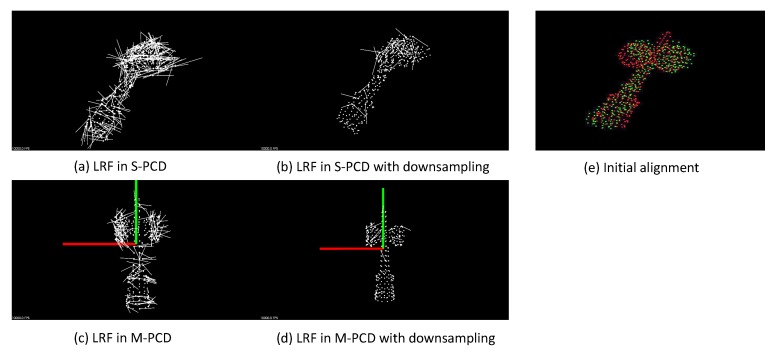
Result of initial matching.

**Table 1 sensors-19-01572-t001:** Specifications of projector.

Resolution (pattern sequence mode)	912×1140
Pattern rate (pre-loaded)	4225 Hz
Brightness	150 lm
Throw ratio	1.2
Focus range	0.5–2 m

**Table 2 sensors-19-01572-t002:** Specifications of camera and lens.

Image	8-bit Monochrome
Resolution	512×512
Frame rate	∼2000 Hz
Image size	1/1.8
Focal length	6 mm
FOV	$57.4^{\circ }\times 44.3^{\circ }$

**Table 3 sensors-19-01572-t003:** Specifications in two axial directions.

	Pan	Tilt
Type of servo motor	Yaskawa SGMAS-06A	Yaskawa SGMAS-01A
Rated output [W]	600	100
Max torque [Nm]	5.73	0.9555
Max speed [rpm]	6000	
Reduction ratio	4.2	

**Table 4 sensors-19-01572-t004:** 4-bit Gray code.

Digit	Gray Code
0	0000
1	0001
2	0011
3	0010
4	0110
5	0111
6	0101
7	0100
8	1100
9	1101
10	1111
11	1110
12	1010
13	1011
14	1001

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
