# Peer review of "Development of an Active High-Speed 3-D Vision System"

_sensors, 2019, doi:10.3390/s19071572_

Reviewer 1 Report

 The paper proposes a high-speed 3-D sensing system with active target-tracking. The system consists of a high-speed camera head and a high-speed projector, which are mounted on a two-axis active vision system. A coded pattern projection is used to achieve a rate of 500 fps. The measurement range is increased because of the active tracking and of the accurately the shape of the target is observed, even when it moves quickly, performing a Realtime model matching is achieved.
The method is well structured and exposed. The system is tested in real conditions, showing good performance. A link to a demonstrator video is provided.

To improve or better explain:

- Layout of figures 7-11 and 12-15. Please try to Insert the figures with the text.
- Please, compare your results with other previous visual tracking results in literature.
- Have you tested your system using different shapes and size of objects?
- What happened if you loose the object in the image? Can the tracking be recovered?
- The successful of your method is due to the quality of the camera or to the pattern projection technique? Have you compared results?

Author Response

We would like to thank the reviewer for careful and thorough reading of this manuscript and for the thoughtful comments and constructive suggestions, which help to improve the quality of this manuscript. Our response is written in the attached file.

Reviewer 2 Report

The authors present an active high-speed 3D vision system for object tracking based on structured light, image-based visual servoing and point cloud model matching. Overall, the paper presents an interesting system with supporting experimentation, however some parts of the paper are not clear and need to be improved.

The writing should be improved. In this regard, I recommend the authors to revise their text with an English native speaker.

The research seems to be strongly based on multiple authors’ previous works. Although these works are introduced in Section 2, it is difficult to determine the contribution of this paper. In this direction, a paragraph clarifying the contribution of this specific paper will be of great help.

Some parts of the paper are rather confusing and require a proper clarification:

1.  The authors claim that a “3-D measurement at a rate of 500fps is achieved” is confusing. In detail, in section 3, the authors explain that the system synchronizes a projector and a camera devices working at 1000fps, and, in section 3.3, they claim that the “depth image is generated at 500 Hz”. Later on, in section 4.3, it is explained that “the measurement rate is 1kHz for every single pattern” and “18 times of pattern recognition is needed to generate one depth image” explaining that “the delay of depth sensing become 18ms’”.

The 18 patterns and 18ms delay defines a frequency of near 56Hz and seems contradicting the previous claims. In this direction, the authors should clarify this part and how the claimed 500Hz number is defined.

2. In section 5.2, the position target r_s is defined on the camera coordinates Sigma_s, where x_s is defined as the depth value, as represented in Figure 3. It is not clear how this coordinate frame has been obtained from the camera frame Sigma_c, represented in Figure 5.

3. In section 6.1, an initial alignment defined by matching Local Reference Frames(LRF) computed from points with “large curvature” is explained. However, the authors did not explain how those points were selected from the normalized curvatures, how the actual matching was performed between LRF and how a matching was considered correct.

4. In Experiments, it is not clear if the authors use the model matching for the experiment of section 7.1. In this direction, the authors only mention the “3-D trajectory of the center of gravity of the target” and also “the time to calculate the depth value was 1.5 ms”.

Other points:

- In section 4.1, the first pattern, shown in Figure 5(a), is fully black, therefore seems that it doesn’t provide any spatial information. What is its function?

- In equation 7, summation subscripts i and j are not used.

- In equation 15, a minus is missing in front of the focal length located at the second row of the right matrix?

- Under equation 15, y_x should be y_s.

-In section 7.2, regarding model matching, authors wrote that the “The number of S-PCD at this time was about 100, and the total processing time combined with the acquisition of 3D data was about 0.9~1.1s”. In this regard, is not clear which time the authors mean and from where the number in seconds is obtained (perhaps should be ms?).

- Figure 7(a) do not show the X-axis.

Author Response

(The authors gave the same response as above.)

Reviewer 3 Report

The Authors presented a fast 3D Vision system based on fringe patterns used for 3D scanning and additional tracking.

The paper is generally well written and the results are interesting. Nevertheless, there are some related paper missing in bibliography and therefore I would encourage the Authors to compare their approach with some other similar systems presented by various researchers.

DOIs of some such papers are: 10.1007/s11263-012-0554-3 , 10.1364/OE.22.001287 ,  10.1109/MMAR.2012.6347906 , 10.1364/AO.55.004395 , 10.1364/OE.22.026752 , and 10.1364/OE.26.001474.

It would also be interesting to include the additional analysis of the computation time for the consecutive steps of the proposed method (e.g. the ICP which could be potentially replaced by another method even with lower accuracy).

Author Response

We would like to thank the reviewer for careful and thorough reading of this manuscript and for the thoughtful comments and constructive suggestions, which help to improve the quality of this manuscript. Our response is written in the attached file.

Round  2

Reviewer 2 Report

The revision has properly addressed to most questions raised in the previosu review comments. The current manuscript is at a good condition for publication.